# Behavior Cloning from Suboptimal Demonstrations with Robust World Models

## Abstract

Recent advances in behavior cloning and generative modeling of manipulation behaviors have shown promising results in learning complex multi-modal behavior distributions. However, a common limitation for all behavior cloning methods has been the challenge of acquiring high-quality training data. Existing state-of-the-art methods for policy learning face significant limitations when expert demonstrations are low quality, and often require the filtering or reweighting of failed or noisy demonstrations. To address this challenge, we propose an efficient offline reinforcement learning framework which utilizes an implicit world model to regularize a behavior cloning policy via predicted future returns. Our approach, Robust Imitation with a Critic (RIC), utilizes a critic-regularized imitation learning objective to incorporate both successful and failed demonstrations, steering imitation learning towards better trajectories via a conservative critic. Our method improves on prior works by accelerating the quality of learned policies by as much as 20% in the presence of suboptimal expert training data. Our simulated experiments consider different types of data suboptimality, including rollouts from a poor demonstrator policy and biased action perturbations from controller error. We empirically evaluate different algorithmic choices for RIC, including comparisons of (1) offline reinforcement learning and behavior cloning, (2) critic guidance via an implicit world-model and a conservative critic estimate, and (3) different behavior cloning methods, including token and diffusion-based architectures.

## 1 Introduction

Behavior cloning is a highly effective method for offline learning of dexterous manipulation policies, especially with the development of generative models such as Diffusion policies [3] and VQ-BeT [4] which can learn multi-modal behavior distributions. Compared to Reinforcement Learning (RL), behavior cloning has the advantage that it does not require a hand-engineered reward function or autonomous interaction with the environment; however, it does assume access to optimal expert demonstrations. Several studies demonstrate that learned dexterous manipulation policies have significantly higher success rates when training data is collected by an expert versus a non-expert demonstrator [5; 6; 7; 8]. Mixed-quality demonstrations are an inevitable challenge in imitation learning, since the quality of demonstrations can vary depending on operator skill, effort, and tiredness, as well as ambient factors such as controller calibration and noise. Additionally, even if the operator is an expert, noise in the sensors used for data collection (such as in proprioception) can also lead to suboptimal demonstrations.

A straightforward approach for dealing with suboptimal data is to simply filter out or reweight suboptimal demonstrations so they contribute less to the learned policy. For instance, Wu et al. [9] and Xu et al. [10] learn discriminators that can be used to identify suboptimal trajectories and reweight them during training. The drawbacks of filtering or downweighting suboptimal trajectories is that we are ignoring the inherent knowledge stored in these trajectories on how not to perform the task and to what extent these actions are suboptimal [11]. This approach also reduces data diversity and therefore robustness in the learned policy.

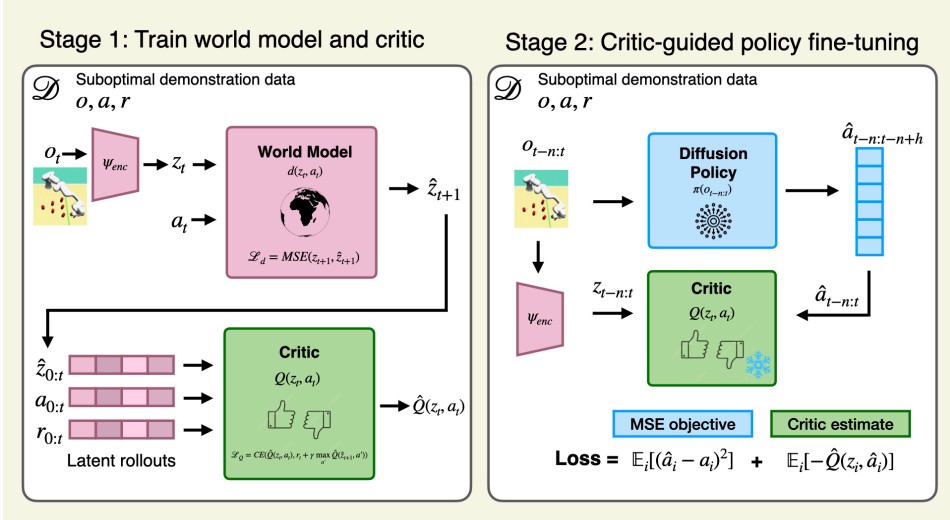

Figure 1: **Robust Imitation with a Critic Overview** The first stage of training involves training an observation encoder, world model, and critic, following the algorithm of TDMPC2 [1]. The world model learns the latent environment dynamics, which helps generate latent environment rollouts that are used to train the critic. Once the critic is trained, the critic is frozen and the second stage of training integrates the critic value estimate into the behavior cloning policy objective. We also explore a variation of our method that uses a conservative IQL critic [2] as opposed to a TDMPC2 critic, in which case we skip the training of the world model.

In contrast to behavior cloning, offline-RL methods such as TDMPC2 [12] and CQL [13] have demonstrated successful performance despite learning from datasets containing suboptimal demonstrations. As proposed in [14], this is in part by stitching together sub-optimal chunks of a trajectory. These methods utilize both optimal and suboptimal data to learn a value function that can guide the policy away from unfavorable actions. However, offline RL is less data efficient than behavior cloning, and it relies on an accurate reward function which must be either hand-engineered or learned via inverse RL [15]. Additionally, traditional offline RL methods are often outpaced by state of the art behavior cloning models like VQ-BeT or diffusion policies if the degree of suboptimality in the dataset is not too large [11]. In this work, we combine the advantages of offline RL and behavior cloning by learning from the entire dataset, including suboptimal trajectories, while retaining the data efficiency of state of the art models used by behavior cloning. We assume that we have access to only a fixed offline dataset, which is common in many settings where it is expensive or infeasible to collect new demonstrations. Our approach uses a learned TDMPC2 world-model to generate latent trajectories that aid in training a conservative offline critic [1]. We then use this learned critic to guide the training of a diffusion policy on suboptimal trajectories [3].

We evaluate our approach on dexterous manipulation tasks to see if we can improve behavior cloning performance in the presence of suboptimal data. Specifically, we evaluate on the PushT task [3], and D3IL stacking and sorting [16], all of which are challenging dexterous tasks with multi-modal expert behavior distributions. We also create synthetic datasets that simulate different sources of suboptimality and show that our approach experiences less performance degradation on these datasets compared to standard behavior cloning methods.

## 2 RELATED WORK

The primary methods for learning policies from partially suboptimal offline datasets are behavior cloning and offline reinforcement learning. This section reviews key contributions in these areas, highlighting advancements in multi-modal policy learning, handling suboptimal

demonstrations, and integrating value-based methods to enhance policy robustness when suboptimal data is present.

## 2.1 BEHAVIOR CLONING

Behavior Cloning (BC) has been a foundational method for policy learning, primarily focusing on directly mimicking expert actions from demonstration data. Recent advancements have emphasized the importance of multimodal policy learning to capture the diverse strategies that experts may employ to accomplish tasks. For instance, the Diffusion Policy framework (3) leverages diffusion models to effectively handle the variability in demonstrations, and BeT (17), and VQ-BeT (4) discretize the action space to better predict high dimensional actions. Behavior cloning has achieved impressive results in dexterous manipulation tasks, but faces challenges when the training dataset includes suboptimal trajectories (5; 6; 7; 8). The issue arises because behavior cloning treats all trajectories equally, regardless of their quality, leading the model to learn behaviors that might not be optimal.

## 2.2 OFFLINE REINFORCEMENT LEARNING

Offline Reinforcement Learning (Offline RL) seeks to learn optimal policies from previously collected non-expert datasets. These methods are able to learn from the entire training set, including suboptimal data, by learning to avoid low value actions and prioritize high value actions. Recent offline RL methods, such as Conservative Q-Learning (13), and its variants like Calibrated Q-Learning (18), focus on minimizing the discrepancy between the learned policy and the behavior policy that generated the dataset, thereby ensuring policy stability and reducing the chance of encountering out of distribution states. In contrast, Implicit Q-Learning (2) enables improving over the behavior policy that collected the dataset, while at the same time staying in distribution to the training data. Model-based approaches like TDMPC2 (12) learn an implicit world model using latent observation representations and use this world model to guide policy and critic learning.

Offline RL methods effectively handle suboptimal trajectories, but are less data-efficient and more unstable than BC due to the need to estimate value functions for all actions. Additionally, their performance may be limited when the reward functions are sparse or poorly defined. While offline RL excels in noise and suboptimal data robustness, state-of-the-art BC methods like VQ-BeT and Diffusion Policy often outperform it in policy performance, especially when the degree of suboptimality in the dataset is not too large (11). Nonetheless, the learned critics remain valuable for guiding policy learning away from detrimental actions.

Recent efforts have also explored the integration of value-based offline RL methods with behavior cloning to maintain data efficiency while enhancing policy robustness. For example, AWAC (19) trains an actor-critic framework, and incorporates a term that constrains the actor to maximize the likelihood of the behavior policy while biasing towards high-advantage actions. QVPO (20) employs a similar approach to train a diffusion policy and bias towards high reward actions, but in an online setting. In contrast to these approaches, our approach combines behavior cloning with value-based methods by training the critic on the offline data first, and then running a second round of training integrating the critic and behavior cloning losses. This ensures the critic's reliability before guiding policy training.

## 2.3 ROBUSTNESS TO DATA CORRUPTION AND SUBOPTIMAL DEMONSTRATIONS

Robustness to data corruption and the ability to learn from suboptimal demonstrations are critical challenges in offline RL and behavior cloning. Behavior cloning approaches are more data efficient than offline RL, since they are simply trying to copy an optimal set of actions. However, they do not naturally account for suboptimal data, and attempts to reweight suboptimal data filters out useful information (9; 10).

On the other hand, offline RL methods naturally handle suboptimal trajectories by guiding learned policies away from low value actions. They also are better able to handle noise in dataset observations, actions, and rewards (21). However, these methods rely on the quality of the reward functions and are far less data efficient than behavior cloning.

Our approach of combining behavior cloning with a critic loss offers a promising way to address these limitations. The critic loss allows the model to evaluate and incorporate the quality of actions directly into the learning process, even for suboptimal trajectories. By integrating the strengths of behavior cloning (efficient learning from trajectories) with those of offline RL (leveraging value estimates), this hybrid approach can make better use of diverse datasets, improve robustness to noisy reward signals, and yield more reliable performance across a range of tasks. Compared to similar methods like AWAC [19] and QVPO [20], our approach first trains a critic using TDMPC2 [12], ensuring critic robustness before applying the critic to the behavior cloning loss. Additionally, our method works with a simple success/failure reward function, since the critic is used only to guide behavior cloning rather than estimate an exact value function.

## 3 METHOD

---

**Algorithm 1** RIC with TD-M(PC)$^2$-based Critic and World Model Training

---

1: **Stage 1: TD-M(PC)$^2$ World Model and Critic Training**
2: **repeat**
3:     Sample a mini-batch $\{(o_t, a_t, r_t, o_{t+1}, \mu_t)\}$ from $\mathcal{D}$
4:     Encode $z_t = h_\theta(o_t)$, $z_{t+1} = h_\theta(o_{t+1})$
5:     Predict reward $\hat{r}_t = R_\psi(z_t, a_t)$ and next latent $z'_{t+1} = d_\psi(z_t, a_t)$
6:     Compute TD target: $y_t = r_t + \gamma \, \mathbb{E}_{a' \sim \pi(\cdot|z_{t+1})} [Q_{\phi'}(z_{t+1}, a')]$
7:     Update critic by minimizing: $\mathcal{L}_{\text{critic}}(\phi) = CE\left(Q_\phi(z_t, a_t), y_t\right)$
8:     Update world model and encoder using consistency, reward, and value losses as in TD-M(PC)$^2$
9: **until** converged or max epochs reached
10: **Fix the critic and world model parameters:** $\theta, \phi$
11: **Stage 2: Critic-Guided Diffusion Policy Training**
12: **repeat**
13:     Sample a mini-batch $\{(o_t, a_t)\}$ from $\mathcal{D}$
14:     Add noise $\epsilon$ to actions at a chosen diffusion step $k$: $\tilde{a}_t^{(k)} = a_t + \sigma_k \epsilon$ *(diffusion forward process)*
15:     For each $k$, predict the noise or score function $\hat{\epsilon}t^{(k)} = \pi\psi(\tilde{a}_t^{(k)}, o_t, \sigma_k)$ *(denoising step, conditioned on noise level)*          *(denoising)*
16:     Recovered actions $\hat{a}_t = \tilde{a}_t^{(0)}$ by iteratively denoising from $k = K$ to $k = 0$
17:     Encode $z_t = h_\theta(o_t)$, estimate $Q_\phi(z_t, \hat{a}_t)$
18:     Compute behavior cloning loss: $\mathcal{L}_{BC} = \|\hat{\epsilon}_t - \epsilon\|^2$
19:     Compute critic regularization: $\mathcal{L}_{RL} = -Q_\phi(z_t, \hat{a}_t)$
20:     Update $\psi$ by minimizing $\mathcal{L}_{RIC}(\psi) = \mathcal{L}_{BC} + \alpha \, \mathcal{L}_{RL}$
21:         *($\alpha$ balances BC vs. RL losses)*
22: **until** converged or max epochs reached
23: **Output:** Critic-guided diffusion policy $\pi_\psi$

---

### 3.1 CRITIC TRAINING USING IMPLICIT WORLD MODELS

We utilize TDMPC2, an offline RL algorithm, to train a critic from an offline dataset, though RIC can work with other methods for critic training as well (see Experiment 3 for results with an IQL [2] critic). We chose TDMPC2 over other algorithms like CQL [13] due to its exceptional robustness. TDMPC2 has been shown to successfully train on 104 diverse tasks with consistent hyperparameters, highlighting its generality [1].

TDMPC2 learns an implicit, decoder-free world model to represent environment dynamics. The model predicts future latent states $\hat{z}_{t+1}$ and rewards $\hat{r}_t$ from a current latent state $z_t = h_\theta(o_t)$ (encoded from the observation $o_t$) and action $a_t$. The critic $\hat{Q}_\phi(z_t, a_t)$ is trained using latent one-step rollouts predicted by the implicit dynamics model. These rollouts augment the training dataset and improve the Q-function, which is trained to minimize the

cross entropy ($CE$) between the predicted Q-value at timestep $t$, $\hat{Q}_\phi(z_t, a_t)$ and the one-step bootstrapped target q-value, which is $\max_{a'} \hat{Q}_{\bar{\phi}}(\hat{z}_{t+1}, a')$, yielding:

$$\mathcal{L}_{\text{critic}} = \mathbb{E}\left[CE\big(\hat{Q}_\phi(z_t, a_t), \, r_t + \gamma \max_{a'} \hat{Q}_{\bar{\phi}}(\hat{z}_{t+1}, a')\big)\right] \tag{1}$$

where $\gamma$ is the discount factor. For additional details on how the critic is defined, refer to [1].

Latent rollouts use predicted latent states $\hat{z}_{t+1}$ rather than true latent states from the dataset, effectively functioning as data augmentation via predictive future state encodings. While we currently use true rewards and actions from the dataset for training, future work could explore augmenting rollouts with predicted rewards and actions from the learned reward model and TDMPC2 policy.

Notably, training the world model and critic requires only a basic sparse reward function, since the goal of the critic is simply to guide behavior cloning rather than to exactly estimate the value function. This is also the reason that the critic can be trained on rollouts from suboptimal experts that generated the dataset, and its guidance can still effectively support the training of a different policy—the learned diffusion policy.

### 3.2 Critic-Guided Diffusion Training

We use the Diffusion Policy for behavior cloning due to its demonstrated success in dexterous manipulation tasks and its natural ability to model multi-modal behavior distributions [3]. In baseline testing, the Diffusion Policy outperformed alternatives like VQ-BeT [4] and TDMPC2 [12] on the PushT [3] and D3IL [16] tasks.

**Diffusion Policy Overview** The Diffusion policy learns to generate actions by iteratively denoising a set of noisy actions conditioned on observations. At each step, a chunk of actions $a_{t-n:t-n+h}$ and the corresponding observation history $o_{t-n:t}$, where $h > n$, the model predicts a noise-conditioned score function that estimates the gradient of the log probability of the clean action, given the current noisy action $\tilde{a}_{t-n:t-n+h}$:

$$\tilde{a}_{t-n:t-n+h} = a_{t-n:t-n+h} + \epsilon_{t-n:t-n+h} \tag{2}$$

Formally, at each step $k$ of the reverse diffusion process, given a noisy action $\tilde{a}_{t-n:t-n+h}^{(k)}$ and observation history $o_{t-n:t}$, the model predicts the score $\hat{s}_k(\tilde{a}_{t-n:t-n+h}^{(k)}, o_{t-n:t}, \sigma_k)$, where $\sigma_k$ is the noise level at step $k$. The denoising process proceeds over multiple steps, gradually removing noise from an initial random action sample, with the full loss being

$$\mathcal{L}_{\text{Diffusion}} = \mathbb{E}_{k,i}\left[(\hat{\epsilon}_i^{(k)} - \epsilon_i^{(k)})^2\right] \tag{3}$$

Predicted actions are computed as:

$$\hat{a}_{t-n:t-n+h} = \tilde{a}_{t-n:t-n+h} - \hat{\epsilon}_{t-n:t-n+h} \tag{4}$$

**RIC: Critic-Guided Diffusion Policy Training** RIC augments the diffusion policy loss with the TDMPC2 critic's value estimate for the predicted actions. Given an action chunk $a_{t-n:t-n+h}$, the predicted noise $\hat{\epsilon}_{t-n:t-n+h}$, and the predicted actions $\hat{a}_{t-n:t-n+h}$:

$$\mathcal{L}_{\text{RIC}} = \mathbb{E}_i\left[(\hat{\epsilon}_i - \epsilon_i)^2\right] + \alpha \mathbb{E}_i\left[-\hat{Q}(z_i, \hat{a}_i)\right]$$

The second term biases the policy away from low-value actions using the TDMPC2 critic. Because the TDMPC2 critic is robust to noise, it is capable of predicting accurate values even when the dataset contains noisy or suboptimal demonstrations. This not only de-emphasizes suboptimal trajectories but actively helps the policy learn to avoid highly suboptimal actions, thereby improving overall robustness and performance. During training, the critic is fixed, and only the diffusion policy is updated based on this combined loss.

In our implementation, we first train the diffusion policy without critic guidance for the initial half of the training period (i.e., setting the second term of the RIC loss to zero). This allows the policy to establish a strong baseline by focusing solely on imitation learning from the dataset. In the second half of training, we introduce critic-guided distillation to refine the policy further by steering it away from low-value actions. This two-stage approach ensures that the policy first learns generalizable patterns from demonstrations before leveraging the critic's feedback to improve robustness.

### 3.3 Test-Time Policy Execution

At test time, given an observation history $o_{t-n:t}$ and a set of random actions $\tilde{a}_{t-n:t-n+h} \sim \mathcal{N}(0,1)$, the diffusion policy predicts the noise $\hat{\epsilon}(\tilde{a}_{t-n:t-n+h}, o_{t-n:t})$ to subtract from the noisy actions:

$$\hat{a}_{t-n:t-n+h} = \tilde{a}_{t-n:t-n+h} - \hat{\epsilon}_{t-n:t-n+h}$$

The TDMPC2 critic is not used during test time. In practice, both during training and test time we use relative delta actions for each timestep from $a_{t-1}$, making each action roughly centered around mean 0.

## 4 Experiments

| Method | | Diffusion | VQ-BeT | TDMPC2 | IQL | Ours |
|---|---|---|---|---|---|---|
| | | | | Baselines | | |
| Uses data | | $(\mathbf{o}, \mathbf{a})$ | $(\mathbf{o}, \mathbf{a})$ | $(\mathbf{o}, \mathbf{a}, r)$ | $(\mathbf{o}, \mathbf{a}, r)$ | $(\mathbf{o}, \mathbf{a}, r)$ |
| Training Method | | Offline | Offline | Offline + Online | Offline | Offline |
| Domain | Task Name | | | | | |
| D3IL | Stacking (1) | $0.69 \pm 0.02$ | $0.43 \pm 0.05$ | $0.36 \pm 0.11$ | $0.00 \pm 0.00$ | $\mathbf{0.71 \pm 0.06}$ |
| | Sorting (2) | $\mathbf{0.95 \pm 0.01}$ | $0.91 \pm 0.02$ | $0.41 \pm 0.03$ | $0.21 \pm 0.10$ | $0.93 \pm 0.03$ |
| PushT | PushT | $0.91 \pm 0.04$ | $0.75 \pm 0.08$ | $0.08 \pm 0.03$ | $0.11 \pm 0.02$ | $\mathbf{0.94 \pm 0.02}$ |

Table 1: Baseline comparisons on D3IL and PushT tasks using the original datasets. RIC achieves performance comparable to the best baselines, showing that critic guidance does not negatively impact imitation learning performance on predominantly optimal datasets. In two out of three tasks, RIC slightly surpasses the baselines, suggesting that even these datasets can benefit from critic guidance, perhaps due to small amounts of noise present.

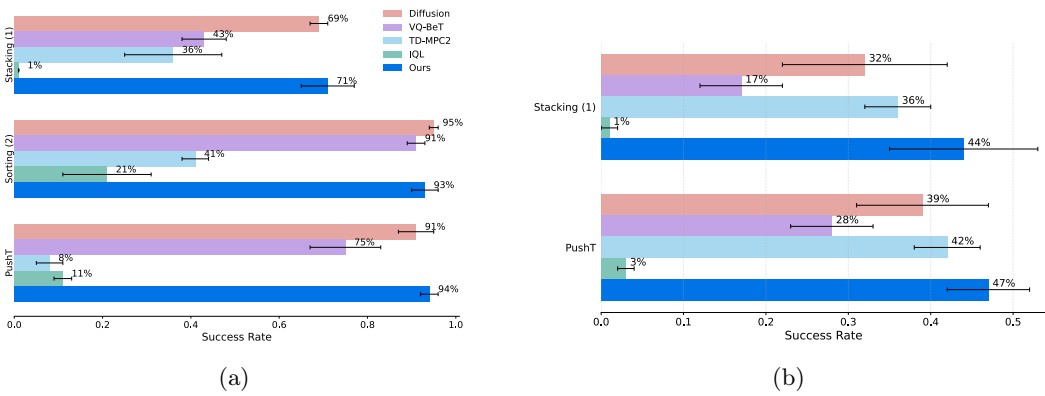

(a)                                                 (b)

Figure 2: Side-by-side comparison showing the drop in per-task performance from (left) best-case (i.e. expert demonstrations) to (right) demonstrations with action noise scenarios. We find that RIC performs comparably to the best policy (Diffusion) when given only expert demonstrations. All of the methods lose performance when action noise is added to the datasets, but in this case RIC significantly outperforms the baselines due to the guidance from critic value estimates. All error bars shown are the standard deviations for 5 random seeds.

|  |  | Baselines | | | | |
| Method | | Diffusion | VQ-BeT | TDMPC2 | IQL | Ours |
| Uses data | | $(\mathbf{o}, \mathbf{a})$ | $(\mathbf{o}, \mathbf{a})$ | $(\mathbf{o}, \mathbf{a}, r)$ | $(\mathbf{o}, \mathbf{a}, r)$ | $(\mathbf{o}, \mathbf{a}, r)$ |
| Training Method | | Offline | Offline | Offline + Online | Offline | Offline |
| Domain | Task Name | | | | | |
| D3IL | Stacking (1) | $0.32 \pm 0.10$ | $0.17 \pm 0.05$ | $0.36 \pm 0.04$ | $0.01 \pm 0.01$ | $\mathbf{0.44 \pm 0.09}$ |
| PushT | PushT | $0.39 \pm 0.08$ | $0.28 \pm 0.05$ | $0.42 \pm 0.04$ | $0.03 \pm 0.01$ | $\mathbf{0.47 \pm 0.05}$ |

Table 2: D3IL and PushT task success rates for different algorithms when demonstrations are perturbed by biased noisy actions. We note that the significant difference in performance between RIC and diffusion is driven by the high success rates of TDMPC2, which is able to still succeed at the task in the presence of noisy actions. This highlights the advantage gained when using the implicit world model for critic estimates over model-free methods when noise contaminates the data, as enough noise-free interactions exist in the data to construct the world model that produces more accurate critic estimates.

|  |  | Baselines | | | | | |
| Method | | Diffusion | Filtered Diffusion | VQ-BeT | TDMPC2 | IQL | Ours |
| Uses data | | $(\mathbf{o}, \mathbf{a})$ | $(\mathbf{o}, \mathbf{a})$ | $(\mathbf{o}, \mathbf{a})$ | $(\mathbf{o}, \mathbf{a}, r)$ | $(\mathbf{o}, \mathbf{a}, r)$ | $(\mathbf{o}, \mathbf{a}, r)$ |
| Training Method | | Offline | Offline | Offline | Offline + Online | Offline | Offline |
| Domain | Task Name | | | | | | |
| D3IL | Sorting (2) | $0.58 \pm 0.00$ | $0.79 \pm 0.02$ | $0.40 \pm 0.03$ | $0.53 \pm 0.04$ | $0.48 \pm 0.15$ | $\mathbf{0.91 \pm 0.05}$ |
| PushT | PushT | $0.55 \pm 0.05$ | $0.64 \pm 0.04$ | $0.37 \pm 0.01$ | $0.61 \pm 0.06$ | $0.11 \pm 0.02$ | $\mathbf{0.85 \pm 0.06}$ |

Table 3: Baseline comparisons on D3IL and PushT tasks with suboptimal demonstrations. RIC significantly outperforms both the behavior cloning and offline RL baselines, including a baseline of a diffusion policy trained on a dataset filtered to only include successful trajectories. Note that for this experiment, we evaluated RIC with an IQL critic instead of a TDMPC2 critic, as IQL value estimates were better when training on datasets containing a significant number of unsuccessful demonstrations, rather than just with noisy trajectories.

We evaluate our approach on a set of dexterous manipulation tasks, including the PushT benchmark introduced by Diffusion Policy [20], and D3IL stacking and sorting [16]. These benchmarks provide offline datasets of demonstrations for training. Notably, the D3IL benchmarks provide simple success/failure sparse reward functions, allowing us to evaluate our approach in a sparse reward scenario. We evaluate first on the original datasets, which themselves contain degrees of suboptimality. For each of the tasks, we then generate synthetic datasets to represent different sources of suboptimality. We consider the following two common sources of suboptimality in demonstrations: (1) a poor demonstrator, which we emulate by using rollouts from a partially trained policy, and (2) action perturbations, which simulate common challenges with noise in teleoperated demonstrations.

Jia et al. [16] establish baselines for various models on the D3IL tasks; for instance, their DDPM-based diffusion policy achieves approximately 90% success on the block sorting task using state-based observations (block and agent positions). Motivated by this success, we also focus on state-based observations in our evaluation for these tasks. For PushT, we evaluate policies using both state-based observations and images.

For each task, we compare Diffusion Policy [20], VQ-BeT [4], TDMPC2 [1], IQL [2] and our proposed approach to provide a comprehensive evaluation spanning behavior cloning and offline RL baselines. To maintain consistency with prior work, we train the baselines for 500 epochs for each task. For RIC, we train the critic for 250k+ steps (for most critics we train for less than 250k steps, but for a few of our experiments the critic required more steps to converge to its lowest value loss). Due to a small critic architecture, critic training is much faster than training the baselines. We then train a diffusion policy for 125k steps, and run an additional 125k steps of fine-tuning with critic distillation. This ensures that the total number of training steps in RIC is comparable to the baselines. Additionally, we train three random seeds per task and evaluate maximum success rates using 50 trials per seed, where each trial involves randomized block or target positions.

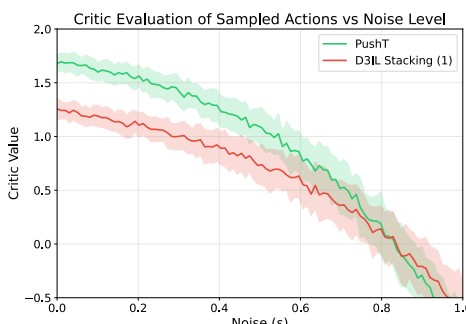

Figure 3: Plotting the predicted TDMPC2 normalized critic value for actions with different levels of noise (with $\gamma = 0.995$). The critic was trained with action noise perturbed demonstrations, but still assigns higher expected values to high quality actions and low values to low quality (noisy) actions. Therefore, the critic gradient is useful for RIC in guiding the policy away from noisy actions.

### 4.1 EXPERIMENT 1: PERFORMANCE ON STANDARD DATASETS

We begin by evaluating performance on the original offline datasets provided for each task. This experiment ensures our approach maintains or improves performance relative to existing baselines. Our results are shown in Table 1. Our experiments showed slightly improved performance on both the D3IL and PushT tasks, indicating that even offline datasets without significant suboptimality can benefit from critic guidance. In this case, the improvement may be due to small amounts of noise present in the datasets that could limit the performance of imitation learning baselines.

### 4.2 EXPERIMENT 2: ROBUSTNESS TO NOISY CONTROL INPUTS

To simulate teleoperation errors, we introduce episode-level action noise into the datasets by adding Gaussian noise uniformly to all actions within an episode. This mimics a consistent operator error throughout the episode, which can occur from a controller miscalibration. The noise magnitude ranges from 2% to 25% of the standard deviation of each action dimension, representing varying levels of teleoperation errors across different action dimensions. Importantly, we keep the original reward labels, assuming that the operator adapted to the miscalibration and still successfully completed the task. We evaluate our method alongside baselines on the D3IL stacking and PushT tasks, presenting results in Table 2. In this instance, the Sorting task is omitted due to the policy being unsuccessful in completing any of the sorting rollouts when noise is applied, to contrast with both the PushT and D3IL-Stacking domains. Additionally, Figure 2 visualizes the performance of the baselines and RIC, both with and without added action noise.

A key observation is the substantial performance gap—up to 44%—between offline RL methods and RIC when action noise is added to the datasets. This discrepancy likely arises because offline RL policies depend on learned Q-values, which can be sensitive to small inaccuracies. At test time, slight errors in Q-value estimates may push the policy out of distribution, where Q-values become highly unreliable. Noisy actions exacerbate this issue by increasing the likelihood of such deviations. In contrast, behavior cloning (BC) policies remain closer to the training distribution since they directly imitate dataset actions. However, when the dataset itself contains noise, BC policies replicate these errors rather than correcting them, as reflected in their performance drop in Figure 2 when action noise is introduced.

In contrast, RIC, which integrates behavior cloning with offline RL, benefits from both approaches. Its imitation learning component helps keep it in distribution, while the critic refines actions by providing corrective feedback. As shown in Figure 3, a trained TDMPC2 critic's estimated value decreases as action noise increases. Despite being trained on noisy data, the critic still supplies meaningful gradients that guide RIC away from poor actions. The critic's robustness to noise likely arises from the TDMPC2-learned world model, which

generates latent rollouts that support learning a reliable value function. Consequently, RIC achieves approximately a 10% improvement in success rate on average over a pure diffusion policy on the noisy datasets.

### 4.3 Experiment 3: Robustness to Suboptimal Demonstrators

To evaluate robustness to suboptimal expert demonstrations, we generate new datasets using a partially trained policy. Specifically, we use diffusion policies trained for 25K steps on each task, resulting in datasets with success rates of 57.5% on D3IL sorting and 57.1% on PushT. These synthetic datasets are kept the same size as the original datasets to ensure a fair comparison. We then evaluate our method and the baselines on these datasets, with results presented in Table 3.

In this experiment, we introduce an additional baseline: a diffusion policy trained on a filtered version of the dataset that retains only successful trajectories. Filtering out failed demonstrations is a common approach for handling suboptimal expert data, and our results confirm that this strategy enhances diffusion policy performance compared to training on the full suboptimal dataset. However, RIC is able to surpass this performance by leveraging information from the entire dataset, including failed trajectories. Specifically, a critic is trained on the full dataset, and imitation learning with critic guidance is then applied to the filtered dataset.

Interestingly, we observed that using a TDMPC2 critic with RIC did not improve diffusion policy performance in this setting. However, replacing it with an IQL [2] critic led to a substantial 12-20% improvement over the best baseline, which was a diffusion policy trained on the filtered dataset. We hypothesize that this is because the TDMPC2 critic may overestimate value estimates, especially when learning from suboptimal data which includes many failed demonstrations, whereas IQL learns a conservative value function that remains more reliable in the presence of a large number of failed trajectories (i.e. if the success rate < 70%). Future research could explore methods such as [22] to enhance the reliability of TDMPC2 critic estimates when dealing with high proportions of unsuccessful trajectories. This would allow us to retain the benefits of the TDMPC2 world model, enabling latent rollouts and improving robustness to noise in the critic distillation.

## 5 Conclusion

We present our method, Robust Imitation with a Critic (RIC), as an approach to mitigate one of the key challenges in behavior cloning: learning with suboptimal demonstration data. By leveraging policy guidance from a model-based critic trained via Offline RL, we show that our method is robust to suboptimal demonstrations, similar to data-quality experiments done in prior work (7; 21). We evaluate RIC by a) adding an episode level action bias to expert demonstrations, and b) using a suboptimal policy to generate demonstration trajectories with partial failures. Empirically, our results demonstrate that hybrid critic-guided behavior cloning outperforms standard behavior cloning methods—even when filtering out all suboptimal demonstrations—by leveraging value-based policy iteration alongside the behavior cloning objective. However, we observe limitations when the demonstration dataset contains a high proportion of failed demonstrations, where conservative Q-learning approaches such as IQL [2] outperform the model-based Offline RL critic we use, TDMPC2 [1]. To address this limitation, future work could focus on developing a model-based critic that mitigates action-value overestimation when used for policy guidance.

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
