## Ablation and Sensitivity Analysis

| Domain | Task | TD-M(PC)$^2$ Critic + RIC (Ours) | TD-M(PC)$^2$ (World Model, no BC) | IQL (No World Model, no BC) | IQL Critic + Ours (Ours, No World Model) |
|---|---|---|---|---|---|
| D3IL | Sorting (2) | $0.78 \pm 0.05$ | $0.53 \pm 0.04$ | $0.48 \pm 0.15$ | $0.91 \pm 0.05$ |
| PushT | PushT | $0.92 \pm 0.07$ | $0.62 \pm 0.06$ | $0.11 \pm 0.02$ | $0.85 \pm 0.06$ |

Table 4: Qualitative ablation results for D3IL and PushT, isolating the contributions of the world model and critic type. Each column shows the mean $\pm$ std success rate for the specified configuration.

| Task | $\alpha = 0.0$ | 0.25 | 0.5 | 1.0 | 2.0 |
|---|---|---|---|---|---|
| **PushT** | $0.55 \pm 0.05$ | $0.63 \pm 0.07$ | $0.74 \pm 0.05$ | $0.85 \pm 0.06$ | $0.85 \pm 0.07$ |

Table 5: Sensitivity analysis of PushT task success rate as a function of the RL loss coefficient $\lambda_{\mathrm{RL}}$. Each column shows the mean $\pm$ std success rate for a different value of $\lambda_{\mathrm{RL}}$.

## A    Experimental Details and Hyperparameters

We detail the core training, network architecture, and algorithm-specific hyperparameters for all evaluated methods and tasks. Tables 6, 7, and 8 summarize the settings used for each algorithm and environment.

Table 6: Core Training Hyperparameters

| Algorithm | |
|---|---|
| **VQBET** | |
| Learning Rate | 1e-4 (main) |
| Training Steps | 2,000,000 |
| Batch Size | 64 |
| **TDMPC$^2$** | |
| Learning Rate | 3e-4 |
| Training Steps | 500,000 |
| Batch Size | 256 |
| **Diffusion** | |
| Learning Rate | 1e-5 |
| Training Steps | 2,000,000 |
| Batch Size | 64 |
| **RIC** | |
| Learning Rate | 1e-4 |
| Training Steps | 2,000,000 |
| Batch Size | 64 |

Table 7: Network Architecture Parameters

| Algorithm | |
|---|---|
| **VQBET** | |
| GPT Layers | 8 |
| GPT Hidden Dim | 512 |
| MLP Hidden Dim | 1024 |
| VQVAE Embedding Dim | 256 |
| **TDMPC$^2$** | |
| State Encoder Hidden | 256 |
| Latent Dim | 512 |
| Task Dim | 96 |
| MLP Dim | 512 |
| Q Ensemble Size | 5 |
| **Diffusion** | |
| Down Dims | 512, 1024, 2048 |
| Diffusion Embed Dim | 128 |
| Kernel Size | 5 |
| **RIC** | |
| Task Dim | 32 |
| MLP Dim | 256 |
| Latent Dim | 256 |
| Image Encoder Hidden | 256 |

Table 8: Algorithm-Specific Parameters

| Algorithm | |
|---|---|
| **VQBET** | |
| N Obs Steps | 5 |
| Action Pred Tokens | 3 |
| Action Chunk Size | 2 |
| Block Size | 500 |
| **TDMPC$^2$ [22]** | |
| Horizon | 10 |
| N Action Steps | 10 |
| N Gaussian Samples | 512 |
| CEM Iterations | 6 |
| N Elites | 64 |
| Reward Coef. | 0.1 |
| Value Coef. | 0.1 |
| Entropy Coef. $(\alpha)$ | 1e-4 |
| Scale theshold $(s)$ | 2.0 |
| Prior Constraint () | 1.0 |
| **Diffusion** | |
| N Obs Steps | 2 |
| Horizon | 8 |
| N Action Steps | 2 |
| Timesteps | 100 |
| Inference Steps | 16 |
| **RIC** | |
| N Obs Steps | 5 |
| Action Pred Tokens | 3 |
| Action Chunk Size | 2 |
| Num Bins | 100 |