# OpenReview forum: "Behavior Cloning from Suboptimal Demonstrations with Robust World Models"
_ICLR.cc/2026/Conference — ICLR 2026 Conference Withdrawn Submission_

### Official Review · Reviewer_Cjrk · 2025-10-29

**Soundness:** 2
**Presentation:** 3
**Contribution:** 2
**Rating:** 2
**Confidence:** 4

**Summary:**

In this paper, to address the challenge that high-quality training data is usually required for existing behavior cloning methods, an offline reinforcement learning framework, namely Robust Imitation with a Critic (RIC), is presented, where an implicit world model and a critic-regularized imitation learning objective are utilized to regularize a behavior cloning policy via predicted future returns and incorporate both successful and failed demonstrations. Simulated experiments consider different types of data suboptimality show the effectiveness and robustness of the proposed approach.

**Strengths:**

1. The presented idea is simple and easy to execute, also the method is effective in some cases.
2. The presented method can learn the policy from the entire dataset, including suboptimal trajectories, while retaining the data efficiency of behavior cloning.
3. Experimental results show that the proposed framework can improve the performance of BC methods even with suboptimal data.

**Weaknesses:**

1. The presented framework is mainly the direct combination of existing techniques, the novelty and contribution are not significant.
2. While experimental results verify the effectiveness and robustness of the proposed framework, theoretical guarantees and analysis for the performance are lacking.
3. True rewards and actions from the dataset are still needed for the training of the world model, while such prior information is not required for many BC-related approaches.
4. As the diffusion process may be time-consuming and computationally expensive, in the experiments, besides the experimental results reported, the time used and the computation cost of different baselines should also be compared and discussed.  Also, more recent baselines can be added for the comparison.

**Questions:**

1. Ablation study to show the impact of the accuracy of the learned world model on the final performance should be conduct.
2. Please refer to the weakness points.

---

### Official Review · Reviewer_5cF6 · 2025-10-29

**Soundness:** 2
**Presentation:** 3
**Contribution:** 2
**Rating:** 4
**Confidence:** 4

**Summary:**

This paper tackles the critical problem of behavior cloning from suboptimal demonstrations. The authors propose **Robust Imitation with a Critic (RIC)**, a two-stage hybrid method. First, a robust offline RL critic (e.g., TDMPC2, IQL) is trained on the full, mixed-quality dataset.
Second, this frozen critic is used as a regularizer to guide the training of a diffusion policy, steering it away from low-value actions while still primarily imitating the data. The method is shown to significantly outperform standard BC and offline RL baselines on manipulation tasks with synthetic suboptimality, such as noisy actions and partially-failed demonstrations.

**Strengths:**

1. **Important Problem**: Handling suboptimal or noisy demonstrations is a well-recognized problem in imitation learning. The paper targets a practically important problem that arises recently in robot learning.

2. **Novelty**: The proposed RIC framework integrates model-based critic guidance into diffusion-based imitation learning is novel.

3. **Strong Empirical Result**: RIC outperforms baseline under both original, noisy, and suboptimal settings.

4. **Presentation**: The paper is well-written and easy to follow. Figures and tables are clear and informative.

**Weaknesses:**

1. **Lack of Theoretical Analysis**:
The paper motivates RIC empirically but provides little theoretical insight into how or why critic regularization guarantees improvement under suboptimal data. Some analysis (e.g., convergence or bias under noisy demonstrations) would strengthen the contribution.

2. **Evaluation Scope**:
The benchmarks are primarily simulation-based. It remains unclear whether the proposed approach scales to real-world robot learning.
Experiments are limited to 3 tasks. More diverse tasks could enhance generality.

3. **Computation Resource Analysis**:
Training involves both the world model, the critic, and the diffusion policy. It would be useful to report computational costs relative to standard diffusion policy training.

4. **Missing World Model**: The paper's title and abstract heavily feature "Robust World Models." However, the world model is not evaluated and formally introduced.

5. **Organization**:
Section 4 starts with figures and tables in a row, making it harder to refer to the figures and tables when reading the text part. Reordering the sequence could help the organization.

**Questions:**

1. Experiments 2 and 3 use "fully suboptimal" datasets (e.g., all noisy or all partially trained). In practical settings, datasets often contain a mix of optimal and suboptimal demonstrations (e.g., 50% expert, 25% noisy, 25% poor). How does RIC perform in such mixed distributions? Have you tested RIC on human-collected datasets?

2. The paper categorizes suboptimality into two sources: poor demonstrator and action perturbations. However, real-world data often includes behaviors like recovery or compliance motions that improve robustness rather than degrade performance. Are these cases considered as suboptimality for RIC? How would RIC handle such “useful suboptimality”?

3. In experiment 2, despite RIC having the best performance on the noisy dataset, the success rate of ~40% is still far from usable. How much suboptimality can RIC handle while keeping a good performance, and how is the performance compared with the baselines? It would be good to show the performance comparison with different levels of suboptimality.

4. The world model in RIC requires the reward signal for training, which does not exist in most of the SOTA world models (e.g., Genie 3).  Could RIC benefit from the advancement of the pre-trained world models?

5. Related works about imitation learning with world modeling [1-5] should be discussed.

[1] Nematollahi et al., "LUMOS: Language-Conditioned Imitation Learning with World Models", ICRA 2025.

[2] Huang et al., "Diffusion Imitation from Observation", NeurIPS 2024.

[3] Kolev et al., "Efficient Imitation Learning with Conservative World Models", L4DC 2024.

[4] Lu, Cong, et al., "Synthetic experience replay", NeurIPS 2023.

[5] Ajay, Anurag, et al., "Is conditional generative modeling all you need for decision-making?", ICLR 2023.

---

### Official Review · Reviewer_w329 · 2025-10-31

**Soundness:** 2
**Presentation:** 2
**Contribution:** 3
**Rating:** 2
**Confidence:** 3

**Summary:**

Training on suboptimal expert demonstration is challenging for Behavior Cloning (BC). The paper proposes Robust Imitation with a Critic (RIC), a method that integrates a critic to predict future returns and guide the behavior cloning policy. This critic-regularized objective allows the model to learn from both successful and failed demonstrations, steering the policy away from low-value actions. In one of the paper's main approaches, this critic is trained with the help of an implicit world model. The experiments show that the proposed method (RIC) outperforms standard BC methods and offline RL methods, especially in settings with noisy actions or datasets containing many suboptimal demonstrations.

**Strengths:**

- The challenge in suboptimal demonstration training of BC is a key problem to its application, making research on this issue valuable.
- The contribution was easy to follow, and the method is intuitive.
- RIC performs well in the experiment settings.

**Weaknesses:**

- Ablation experiments are not enough to justify the choices of method, including the selection of the critic and the selection of whether to pre-train the policy without guidance.
- The proposed method seems to depend on the critic training method a lot, as seen by the switch from TDMPC2 to IQL in Experiment 3. The paper didn’t have a theoretical basis and a detailed discussion, and a comparison of the selection of the critic training method. Reporting the performance of RIC with different critic guidance systematically across all experiments would probably improve the robustness of the paper.
- The paper does not follow the ICLR reference style (uses numeric [1], [2] citations instead of the required author-year format).

**Questions:**

- The notation in Equation (1) and Algorithm 1 is a bit confusing. For instance, Algorithm 1 (line 3) samples $r_t$ from the dataset, but Section 3.1 mentions the model predicts rewards $\hat{r}_t$. Equation (1) and Algorithm 1 (line 6) then use $r_t$ in the TD target. Could the authors please explicitly confirm if $r_t$ in the critic's loss function is the true reward from the dataset, or what is the usage of $\hat{r}_t$? Additionally, please formally define the notation $\bar{\phi}$ used in Equation (1). Is this a target critic network?
- You report that baseline models (like Diffusion Policy) were trained for 500 epochs, while the policy portion of RIC was trained for 125k steps + 125k steps of fine-tuning (250k total steps). To ensure a fair comparison of computational effort, could you please clarify how many total training steps the 500-epoch baseline training corresponds to?
- In Experiment 2, you noted that the Sorting task was omitted because all policies failed when noise was applied. The noise level ranged from 2% to 25%. Do you think introducing a slightly smaller or different range of noise would allow the policies to achieve some success, which would provide a more complete picture of performance degradation on this task?
- In Experiment 3, you evaluate robustness to suboptimal demonstrators on the D3IL Sorting and PushT tasks (Table 3). Could you also report the performance for the D3IL Stacking task in this same suboptimal demonstration setting?

---

### Official Review · Reviewer_Evqe · 2025-10-31

**Soundness:** 3
**Presentation:** 3
**Contribution:** 3
**Rating:** 6
**Confidence:** 3

**Summary:**

This paper proposes a behavior cloning algorithm that can learn from suboptimal demonstrations using a pretrained critic as a regularizer. The authors propose a two stage process: 1), train a world model and critic given (o, a, r, o') data, 2) train a diffusion policy given (o, a) data and use a regularizer from the critic. In experiments, the proposed method does slightly better than Diffusion policy on the stacking D3IL and PushT tasks, and does much better on noisy actions and suboptimal demonstrations.

**Strengths:**

1. Proposes a hybrid method, in between IL and offline RL.
2. The paper reads well and has a clear algorithm. The experiments also tackle different scenarios with noisy actions and suboptimal demonstrations.
3. The results show that RIC has a clear advantage over diffusion policy.

**Weaknesses:**

- For me, the biggest weakness of this paper is that it does not show exactly how how a simple critic regularizer outperforms offline RL. Both RIC and offline RL require reward and next observations in the dataset. Methods such as TD3+BC [1] or AWAC [2] use a model-free actor critic approach and learn both the policy and critic jointly. There are also DICE-based methods [3,4] that learn density ratios to reweight data. I think it would be good to have a discussion comparing these various methods and talk about when RIC would be more useful compared to others. I would imagine that RIC would perform best in medium noise/optimality scenarios compared to other conservative offline RL algorithms can do better in high noise/low optimal data, whereas on the other end, diffusion policy or BC would be best for highly optimal data. Empirical results can also help in this regard.

- Another is the limited number of tested domains. How would RIC do in say D4RL?

References:
1. Fujimoto, S., & Gu, S. S. (2021). A minimalist approach to offline reinforcement learning. Advances in neural information processing systems, 34, 20132-20145.
2. Nair, A., Gupta, A., Dalal, M., & Levine, S. (2020). Awac: Accelerating online reinforcement learning with offline datasets. arXiv preprint arXiv:2006.09359.
3. Nachum, Ofir, et al. "Dualdice: Behavior-agnostic estimation of discounted stationary distribution corrections." Advances in neural information processing systems 32 (2019).
4. Zhang, Ruiyi, et al. "Gendice: Generalized offline estimation of stationary values." arXiv preprint arXiv:2002.09072 (2020).

**Questions:**

1. Can you clarify why the two stage training procedure is necessary? Does the performance rely heavily on the quality of the critic?
2. I didn't quite understand why the overestimation of the Q-values matter in RIC. Since the Q values are directly used for regularization, isn't it just a matter of using a lower alpha?

---

### Note · Authors · 2025-12-04

**Comment:**

Thanks to the reviewers for their critiques and careful consideration of our work. At this time, we think additional work will be needed to fully address the reviewers main concerns and as such are opting to withdraw the work for incorporating additional improvements. We thank the reviewers and AC for their time in reviewing this work.

**Withdrawal Confirmation:**

I have read and agree with the venue's withdrawal policy on behalf of myself and my co-authors.